# Trends and Patterns of Knee Osteoarthritis in China: A Longitudinal Study of 17.7 Million Adults from 2008 to 2017

**DOI:** 10.3390/ijerph18168864

**Published:** 2021-08-23

**Authors:** Hongbo Chen, Junhui Wu, Zijing Wang, Yao Wu, Tao Wu, Yiqun Wu, Mengying Wang, Siyue Wang, Xiaowen Wang, Jiating Wang, Huan Yu, Yonghua Hu, Shaomei Shang

**Affiliations:** 1Department of Epidemiology and Biostatistics, School of Public Health, Peking University, No. 38 Xueyuan Road, Beijing 100191, China; chenhongbo@bjmu.edu.cn (H.C.); jhsophie@163.com (J.W.); wangzijing14@pku.edu.cn (Z.W.); yaowu@pku.edu.cn (Y.W.); twu@bjmu.edu.cn (T.W.); qywu118@163.com (Y.W.); mywang@bjmu.edu.cn (M.W.); siyue.wang@pku.edu.cn (S.W.); wangxw@bjmu.edu.cn (X.W.); jiating@pku.edu.cn (J.W.); yuhh@pku.edu.cn (H.Y.); 2School of Nursing, Peking University, No. 38 Xueyuan Road, Beijing 100191, China; 3Medical Informatics Center, Peking University, No. 38 Xueyuan Road, Beijing 100191, China

**Keywords:** knee osteoarthritis, prevalence, trends and patterns

## Abstract

**Background:** Knee osteoarthritis (KOA) provides many challenges on the healthcare system. However, few studies have reported the epidemiology, particularly in a large population. Our study aimed to estimate the prevalence, incidence, trends, and patterns of diagnosed KOA in China. **Methods:** This was a longitudinal study. We used health insurance claims of 17.7 million adults from 2008–2017 to identify people with KOA. Trends in prevalence and incidence were analyzed using joinpoint regression. **Results:** We identified 2,447,990 people with KOA in Beijing, 60% of which were women. The 10-year average age-standardized prevalence and incidence of KOA was, respectively, 4.6% and 25.2 per 1000 person-years. Prevalence increased with age, surging after 55 years old. The average crude prevalence was 13.2% for people over 55 years old. The prevalence showed an increasing trend from 2008 to 2017, including a period of rapid rise from 2008 to 2011 (*p* < 0.05); the increase in prevalence was greatest in people under 35 years old (*p <* 0.05). **Conclusion:** Our analyses showed that the annual prevalence rate of KOA increased significantly from 2008 to 2017 in China. We need to increase our attention to women and the elderly over 55 years old, and also be alert to the younger trend of incidence of KOA.

## 1. Introduction

Knee osteoarthritis (KOA) is a common chronic degenerative joint disease, which mainly manifests as knee pain, stiffness, functional limitation, and can even lead to disability [1]. KOA affects approximately 250 million people worldwide, and it is estimated that 10.0–16.0% of older adults over 60 years old have symptomatic knee osteoarthritis worldwide [2,3,4,5]. The years lived with disability (YLDs) caused by osteoarthritis ranked tenth in China in 2016 [6], and the growth rate of YLDs due to osteoarthritis ranked fifth in Nordic countries [7]. Knee osteoarthritis was the first leading among osteoarthritis, accounting for 87% YLDs of osteoarthritis. In China, the number of total knee arthroplasty cases in 2018 was 249,000, and the total cost was about 12.948 billion yuan [8]. Therefore, the disease burden caused by knee osteoarthritis is huge.

Many western countries have conducted epidemiological studies of KOA, including large-scale cross-sectional studies and longitudinal cohort studies [9,10,11]. However, due to the heterogeneity of the population and regional differences, the research in western countries may not be suitable for China. In addition, the existing epidemiological studies of KOA in China have had limited resources and have used relatively small sample sizes [12,13], which could not adequately assess the prevalence and incidence of KOA or its trends and patterns in recent years. 

The aim of the present study was to estimate the prevalence, incidence, trends, and patterns of diagnosed KOA from 2008 to 2017 in Beijing, using the large-scale longitudinal data. It is hoped that the results of this study can provide reference for the prevention, prediction, and rehabilitation management of KOA in China.

## 2. Patients and Methods

### 2.1. Data Sources

Data were obtained from the Beijing Medical Claim Data for Employees (BMCDE), which records medical claims for all beneficiaries who have participated in urban employee basic medical insurance programs in Beijing. Due to the low proportion of the rural population and high employment rate, urban employee medical insurance is the main type of medical insurance in Beijing. At the end of 2017, 17.7 million beneficiaries were included in the BMCDE, covering nearly 80% of the resident population of this city. Information recorded in the database includes demographic characteristics (sex and age), medical information (name of the hospital, level of the hospital, dates of visits), disease diagnosis in Chinese and the corresponding International Classification of Diseases, 10th Revision (ICD-10) codes, dates of diagnoses, and cost information. Details of this database have been presented in previous studies [14,15]. The BMCDE only collects reimbursement data and deletes all personal identifiers, so this study was considered to be exempt from ethical approval because we only used encrypted retrospective information. 

### 2.2. Study Design and Participants

This was a longitudinal study. All beneficiaries aged ≥ 18 years old in the BMCDE from 1 January 2008 to 31 December 2017 were included in the study. Beneficiaries with lack of personal information, unclear diagnosis of disease, and secondary obesity were excluded. Information from the same individual could be linked anonymously using an encrypted number.

### 2.3. Definition of Outcomes

According to the principal diagnosis, patients with KOA were extracted by using ICD-10 codes (M17101, M17.961, M24.661, M25.561, and M25.661). The index date was set to the date when KOA was first diagnosed. Patients newly diagnosed with KOA were defined as those who first met the criteria for KOA (no record of KOA in 2006 and 2007).

### 2.4. Statistical Analysis

Prevalence was calculated by dividing the number of people with KOA each year by the average number of beneficiaries that year (beneficiaries are calculated by taking the average number of beneficiaries on 1 January and 31 December of each year). Incidence was calculated by dividing the number of people newly diagnosed with KOA per year by the total person-years of all beneficiaries that year (those who already have KOA have been subtracted from the beneficiaries). The annual prevalence and incidence rates were standardized through age adjustment, using population data for Beijing from the sixth national census.

We used joinpoint regression to analyze trends in prevalence and incidence rates [16]. Joinpoint regression is a common method to investigate epidemiological time trends that has been used in many studies [17]. This model aims to establish piecewise regression according to the time characteristics of disease distribution, dividing time into different intervals (or phases) through several connection points, and conducting trend fitting and optimization for each interval, so as to evaluate the disease change characteristics of different intervals within the whole time range in more detail. The term joinpoint in the model refers to a significant change in the direction or size of the linear trend found by permutation tests [17]. A log-linear regression model was used in each phase:$$\log\left(y\right)=\beta_{0}+\beta_{1}x$$

In the above equation, y refers to the prevalence or incidence at year x. Annual percent Change (APC), Average Annual percent Change (AAPC), and 95%CIs are the main results of joinpoint models, representing the percentage change in prevalence or incidence in a given year compared to the previous year within each interval and over the whole study period, respectively. APC was calculated based on the slope in each phase, using the following formula:$$\mathrm{APC}=\left(\frac{y_{x+1}-y_{x}}{y_{x}}\right)\times 100=\left(e^{\beta_{1}}-1\right)\times 100$$

The parameter calculation method of AAPC was used to calculate the regression coefficient of each phase by weighting the span width w of the segmented interval [18], using the formula:$$\mathrm{AAPC}=\left(e^{\frac{\sum w_{i}\beta_{i}}{\sum w_{i}}}-1\right)\times 100$$

We analyzed differences using *t*-tests for numerical variables and chi-square tests for categorical variables. The *Z*-test was used to analyze the statistical significance of APC and AAPC, with non-significant changes indicating stable trends [18]. A two-tailed *p* < 0.05 was considered statistically significant. All statistical analyses were performed using SAS 9.3 (SAS Institute Inc., Cary, NC, USA) and the Joinpoint Regression Program (Version 4.8.0.1) from the Surveillance Research Program of the US National Cancer Institute.

## 3. Results

### 3.1. Patient Characteristics

Of the 17.7 million participants, 2,447,990 (14%) people with KOA were identified between 2008 and 2017. Most patients were women (60%), 55–64 years old (28%), retired (58%), diagnosed in tertiary hospitals (52%), and lived in the city (76%). The characteristics of patients with KOA are shown in Table 1.

### 3.2. Prevalence and Incidence of KOA

Table 2 shows the annual age-standardized prevalence and incidence of diagnosed KOA in Beijing from 2008 to 2017. Appendix A shows the annual crude rate of KOA in Beijing from 2008 to 2017. 

The 10-year average age-standardized prevalence of KOA in Beijing was 4.6% from 2008 to 2017. The lowest point appeared in 2008, which was 1.6%, and the highest point appeared in 2015, which was 6.5%. The annual age-standardized prevalence of women was higher than that of men between 2010 and 2017. The prevalence of KOA gradually increased with age, and the rate of increase became greater after 55 years old (Figure 1). The average crude prevalence of the decade was 13.2% for people over 55 years old, with 9.4% for males and 18.0% for females.

The 10-year average age-standardized incidence of KOA was 25.2 per 1000 person-years. The lowest point was 15.2 per 1000 person-years in 2008, and the highest point was 33.9 per 1000 person-years in 2014. The 10-year average crude incidence rate for people over 55 years old was 44.2 per 1000 person-years, with 36.3 per 1000 person-years for males and 54.7 per 1000 person-years for females. 

### 3.3. Trends in the Prevalence and Incidence of KOA

Table 3 shows the trends in the age-standardized prevalence and incidence of diagnosed KOA in Beijing from 2008 to 2017. The prevalence increased significantly with an AAPC of 16.8% (*p <* 0.05) during this decade. There were two different trends in prevalence during this period and one joinpoint in 2011. The period from 2008 to 2011 was a phase of rapid increase, with an APC of 48.9% (*p <* 0.05). This increasing trend in prevalence was observed among all age groups, but the AAPC was greater in people under 35 years old (36.6%, *p <* 0.05). The annual crude prevalence of KOA in people under 35 years old from 2008 to 2017 is shown in Figure 2.

The age-standardized incidence of KOA remained stable from 2008 to 2017, with a non-significant AAPC of 1.3% (*p* > 0.05), and no significant joinpoint. Joinpoint regression showed that the AAPC in the incidence of patients aged 18–34 years old was greater than that of patients aged 35–44 years old (22.9% vs. 19.1%). Moreover, there was a joinpoint in 2012 for both of these age groups. The period from 2008 to 2012 was a phase of rapid increase, with an APC of 56.3% (*p <* 0.05) in the 18–34 age group and 59.3% (*p <* 0.05) in the 35–44 age group. 

## 4. Discussion

To the best of our knowledge, the present study is the first study to describe trends and patterns in the prevalence and incidence of knee osteoarthritis in China. Using data from the health insurance claims of 17.7 million people, we found that the 10-year average age-standardized prevalence and incidence rates of KOA were, respectively, 4.6% and 25.2 per 1000 person-years. Furthermore, prevalence increased significantly within a decade. In addition, we also found that the AAPC in the prevalence and incidence during the decade was the largest in the low-age group (18–34 years), indicating that knee OA was diagnosed at younger ages.

Our results showed that the age-standardized prevalence of KOA in people over 18 years old was 4.6%, which is lower than the studies in the United States (7.3%) and Denmark (13.4%) [10,19]. We attribute the difference in prevalence among studies to differences in race, life and work styles, and other variables. Although the prevalence rate is relatively low compared to some western countries, due to China’s large population, more patients will need rehabilitation treatment and even knee replacement in the future, which could be a heavy public health burden for China. Furthermore, the prevalence observed in the present study increased significantly within a decade, which is similar to the change in prevalence observed in other countries, such as the United States and the United Kingdom [9,20]. The reason for this may be the aging of the population and increased risk factors for knee osteoarthritis (such as low physical activity and high body mass index) in recent years [21,22,23]. In addition, with the development of the economy, the improvement of people’s health-seeking behaviors have also led to the increase in disease detection rate, which may also be one of the reasons for the increasing prevalence of knee OA year by year [24].

It is known that the prevalence and incidence of KOA increases with age. Our study showed that the prevalence and incidence of KOA increased significantly after 55 years old, and that the average crude prevalence from 2008 to 2017 of people over 55 years old was 13.2%: 9.4% for males and 18.0% for females. This is consistent with the results of other current studies. A study on the epidemiology of KOA in the United States showed that the prevalence of knee OA increases with every decade of life, with the annual incidence of knee OA being highest between 55 and 64 years of age [10]. A study in India found that the prevalence of KOA among those who under 50, 50–60, 60–70, and over 70 years old increased sequentially, which is 19.2%, 30.7%, 39.7%, and 54.1%, respectively [25]. The reasons that aging promotes the occurrence of KOA may include the followings [26,27,28]: (i) the cell functions and properties of articular cartilage change with age and it responds differently to cytokines and growth factors; (ii) the articular cartilage secretes less synovial fluid with age, which reduces lubrication of the joints; (iii) muscle strength is reduced with age, so it is difficult to support the surrounding articular cartilage, thereby accelerating cartilage wear; and (iv) after menopause, changes in hormone levels can cause bone hyperplasia and accelerate the onset and progression of osteoarthritis. However, in our study, the prevalence and incidence of KOA in people over 85 years old were lower than they were in the 55–64 and 65–74 age groups. The explanation for this may be related to an increased comorbidity rate and a decreased rate in medical visits due to KOA in this group of people. Another possible explanation for this phenomenon is survivor bias, which allows relatively healthy people to survive to the oldest age group.

In addition, the present study has found the AAPC in the prevalence of knee OA was greatest in the age group under 35 years old. We should be alert to the phenomenon that the rate of KOA is increasing among young people, not only in older adults. In recent years, many chronic diseases have shown a trend of younger disease. Andersson mentioned in the review that [29], in the past 2 decades, a high prevalence of risk factors for cardiovascular disease, such as obesity, physical inactivity, and poor diet, has been observed among young individuals living in developed countries. This leads to a faster increase in the prevalence of cardiovascular disease among young people than among the elderly over 50 years old. Similarly, Wang’s research found that a trend for younger age at type 2 diabetes diagnosis in Beijing [30]. The rejuvenation of chronic diseases will bring a heavy burden of disease, economic, and public health to the country. Therefore, we should pay attention to the health education of young people, cultivate good living habits, reduce the incidence of risk factors, and ultimately control the trend of disease rejuvenation.

Previous studies have shown that the prevalence and incidence of KOA are higher in women than in men [25,31,32], which is consistent with the results of the present study. This phenomenon may be attributed to differences in hormone levels, muscle strength, and health-seeking behavior between the genders [28,33,34]. Estrogen levels in postmenopausal women are significantly lower than premenopausal and male levels, which may affect cartilage metabolism and change the mechanical environment of joints [35]. Moreover, men and women have different sensitivity and tolerance to disease, and women are more likely to seek timely medical treatment than men are [36]. This suggests that men may be less likely to be diagnosed with osteoarthritis, which is reflected in the lower prevalence and incidence among men in this study.

This study has certain strengths. Currently, there is a lack of epidemiological studies on large samples of people with KOA in China. We used data from the health insurance claims of 17.7 million people to estimate the prevalence and incidence of KOA in Beijing in the past 10 years, and determined the prevalence trend for KOA. These findings about KOA could provide valuable evidence for other developing cities in China, and even in cities in other countries in the future. This evidence further clarifies the public health burden of KOA and facilitates the formulation of policies related to prevention, prediction, and rehabilitation management of KOA. Moreover, the estimates of prevalence and incidence were based on a dynamic population, which is closer to real-world population changes.

The limitations of this study include the following. Firstly, since the data were based on hospital visits, we could not obtain information about people who suffered from KOA but did not see a doctor. This may result in an underestimation of prevalence and incidence rates. Secondly, although the BMCDE database covers more than 80% of the resident population of Beijing, we could not obtain information about some immigrants in this city, which may cause selection bias. Thirdly, like many previous studies that have used a large medical database for the purpose of management, we used the ICD code to determine whether a person actually has KOA, but it is difficult to strictly control the diagnostic process, which may lead to bias. Finally, because the BMCDE includes only claims data, information on socioeconomic status, health behaviors, and treatment effects were not available. Therefore, we could not do a more detailed analysis, such as the identification of risk factors.

## 5. Conclusions

Using the large-scale longitudinal data of 17.7 million adults in China, we observed that the average age-standardized prevalence of diagnosed KOA for people over 18 years old was 4.6%, and the crude prevalence rate for people over 55 years old was 13.2%. There were more female patients than men. Our analyses showed that the annual prevalence rate increased significantly from 2008 to 2017, and that KOA was being diagnosed at younger ages. Therefore, we need to increase our attention to women and the elderly over 55 years old, and be alert to the younger trend of incidence of KOA at the same time.

## Figures and Tables

**Figure 1 ijerph-18-08864-f001:**
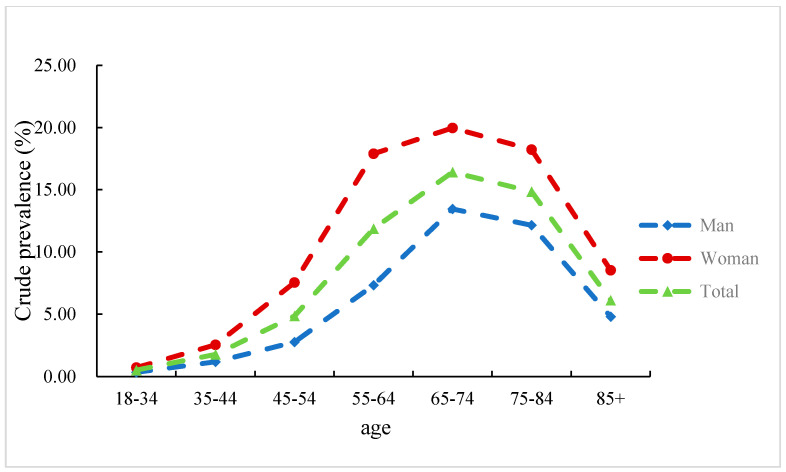
Average crude prevalence of knee osteoarthritis for all age groups in Beijing, China, 2008–2017 (%).

**Figure 2 ijerph-18-08864-f002:**
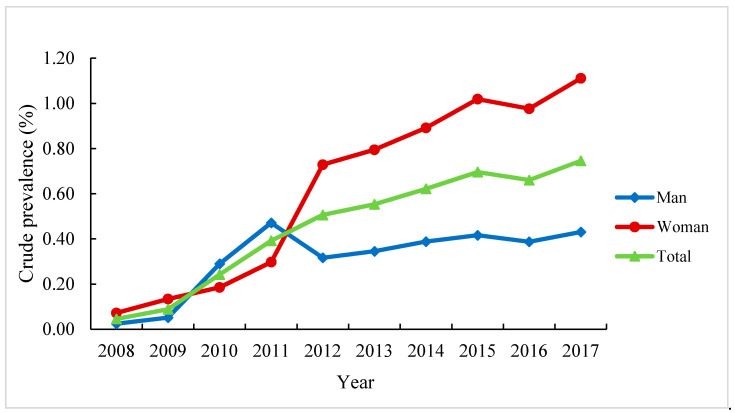
Annual crude prevalence of knee osteoarthritis in people under 35 years old in Beijing from 2008 to 2017 (%).

**Table 1 ijerph-18-08864-t001:** Basic characteristics of people with knee osteoarthritis in Beijing, China, 2008–2017.

Characteristics	Overall (n = 2,447,990)	Men (n = 984,823)	Women (n = 1,463,167)	*p*-Value ^‡^
Level of hospital, %				
Primary	21	22	21	<0.001
Secondary	27	28	26	
Tertiary	52	50	53	
Age group *, %				
18–34 years	9	8	11	<0.001
35–44 years	12	11	13	
45–54 years	21	17	24	
55–64 years	28	29	27	
65–74 years	17	20	14	
65–84 years	11	13	9	
≥85 years	2	2	2	
Area, %				
Urban	76	73	78	<0.001
Rural	24	27	22	
Working state, %				
On work	42	47	38	<0.001
Retired	58	53	62	

Note: * Age group by age at diagnosis; ^‡^
*p*-value for sex differences in the characteristics of people with knee osteoarthritis.

**Table 2 ijerph-18-08864-t002:** Age-standardized prevalence and incidence of diagnosed knee osteoarthritis in Beijing, China, 2008–2017.

	2008	2009	2010	2011	2012	2013	2014	2015	2016	2017
Prevalence (%, 95% CI)										
Overall	1.567 (1.566, 1.568)	2.124 (2.123, 2.125)	3.412 (3.411, 3.413)	4.499 (4.498, 4.500)	5.054 (5.053, 5.055)	5.007 (5.006, 5.008)	6.040 (6.038, 6.041)	6.480 (6.479, 6.481)	5.855 (5.854, 5.856)	5.935 (5.933, 5.936)
Men	1.867 (1.866, 1.868)	2.489 (2.487, 2.490)	2.100 (2.098, 2.101)	2.827 (2.826, 2.829)	2.932 (2.930, 2.933)	2.962 (2.962, 2.964)	3.751 (3.749, 3.752)	4.106 (4.104, 4.108)	3.737 (3.736, 3.739)	3.842 (3.840, 4.843)
Women	1.262 (1.260, 1.263)	1.752 (1.751, 1.753)	4.752 (4.751, 4.754)	6.206 (6.204, 6.207)	7.222 (7.219, 7.224)	7.095 (7.093, 7.098)	8.377 (8.374, 8.379)	8.905 (8.902, 8.907)	8.019 (8.017, 8.020)	8.072 (8.069, 8.073)
Incidence (per 1000 person-years, 95% CI)										
Overall	15.21 (15.20, 15.22)	20.73 (20.72, 20.74)	22.45 (22.44, 22.46)	30.45 (20.72, 30.46)	31.21 (31.20, 31.22)	26.92 (26.91, 26.94)	33.85 (33.84, 33.86)	27.95 (27.94, 27.96)	22.60 (22.59, 22.61)	20.60 (20.59, 20.61)
Men	8.10 (8.09, 8.11)	11.66 (11.65, 11.67)	13.15 (13.14, 13.16)	18.15 (18.14, 18.16)	18.96 (18.95, 18.98)	17.24 (17.23, 17.25)	23.05 (23.04, 23.06)	19.71 (19.69, 19.72)	16.14 (16.13, 16.15)	15.01 (15.00, 15.02)
Women	22.48 (22.47, 22.49)	29.99 (29.98, 30.00)	31.93 (31.92, 31.95)	43.00 (42.98, 43.02)	43.73 (43.72, 43.75)	36.81 (36.79, 36.83)	44.89 (44.87, 44.91)	36.36 (36.35, 36.38)	29.19 (29.18, 29.21)	26.31 (26.30, 26.32)

Note: Population data for Beijing from China’s 6th national census were used to standardize prevalence and incidence.

**Table 3 ijerph-18-08864-t003:** Changes in the age-standardized prevalence and incidence of diagnosed knee osteoarthritis in Beijing, China, 2008–2017 (%).

		Trend 1		Trend 2		Average Annual Percentage Change (95% CI)
		Period	Annual Percentage Change (95% CI)	Period	Annual Percentage Change (95% CI)	
Prevalence	Overall	2008–2011	48.9 (8.5, 91.1) *	2011–2017	4.7 (−0.6, 10.0)	16.8 (8.0, 25.6) *
	Gender					
	Men	–	–	–	–	7.7 (4.7, 10.9) *
	Women	2008–2011	79.3 (20.0, 158.3) *	2011–2017	3.5 (–1.7, 9.3)	24.0 (12.3, 36.1) *
	Age group, years					
	18–34	2008–2011	113.5 (54.3, 189.5) *	2011–2017	9.4 (5.9, 13.0) *	36.6 (25.8, 47.2) *
	35–44	2008–2012	68.1 (33.0, 115.2) *	2012–2017	4.3 (−1.7, 10.4)	29.5 (18.7, 40.1) *
	≥45	2008–2011	49.1 (11.8, 90.0) *	2011–2017	3.8 (−1.0, 9.1)	16.3 (7.8, 25.3) *
Incidence	Overall	–	–	–	–	1.3 (–5.0, 7.4)
	Gender					
	Men	–	–	–	–	4.3 (–2.5, 11.5)
	Women	–	–	–	–	0.2(–6.1, 6.9)
	Age group, years					
	18–34	2008–2012	56.3 (26.4, 88.7) *	2012–2017	2.6 (–3.0, 8.5)	22.9 (14.3, 31.5) *
	35–44	2008–2012	59.3 (17.7, 103.5) *	2012–2017	−3.6 (−11.5, 4.8)	19.1 (7.6, 32.9) *
	≥45	2008–2012	17.5(−1.6, 37.1)	2012–2017	−10.7 (−19.5, 0)	0.8(–6.1, 8.5)

Note: 95% CI, 95% confidence interval. * *p* < 0.05 was considered statistically significant.

## Data Availability

Not applicable.

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
