# Peer review of "Trends and Patterns of Knee Osteoarthritis in China: A Longitudinal Study of 17.7 Million Adults from 2008 to 2017"

_ijerph, 2021, doi:10.3390/ijerph18168864_

Round 1

Reviewer 1 Report

Manuscript ID: ijerph-1326742

Manuscript Title: Trends and patterns of knee osteoarthritis in China: a longitudinal
study of 17.7 million adults from 2008 to 2017

Comments:

In the manuscript, Chen et al mainly investigated the tendency of knee OA in Beijing in China. They indicated that knee OA increased significantly from 2008 to 2017 in China, and there were more women OA patients than men. Moreover, the incident of OA is shown to be more and more popular among young ages. Overall, the finding and analysis from the study is consistent with recent studies related with OA.

Here are some comments and suggestions.

  • For the knee OA, there is different stages of OA in the clinics. Could the data also provide the stages of OA for the patients when they were diagnosed in the hospital? The early diagnosis of OA is one of complicated issues in the research for the clinic study. And it could help us better understand how to prevent OA development at the early stage.

  • OA pain has drawn more and more attention in recent studies. Did the author analyze the drug for pain relief related to knee OA development from the database? Could it work? And how is the efficacy?

  • Obesity is a risk for knee OA. For the analysis, did the author consider the BMI? And the relationship between BMI and OA?

Reviewer 2 Report

This is a longitudinal study that analyzes a large number of patient data.

Major:

1.  In the diagnosis of osteoarthritis, it is very important to exclude other diseases and to refer to the age. For this reason, the American college of rheumatology classification criteria for osteoarthritis of knee specifies age in the diagnosis criteria.

In this study, it was said that the prevalence of patients under the age of 35 is increasing. The authors claimed it is caused by increased tendency of chronic diseases in young people.
However, I think it is also necessary to consider whether the cause of this is from failed differential diagnosis of other diseases or overestimation due to ambiguity of diagnostic criteria. Since it is difficult to control the diagnostic criteria for osteoarthritis in the design of this study, it is likely that this has occurred.

2. It seems necessary to clarify the definition of the word "Incidence". Is "the time of initial diagnosis" the definition of incidence?

Minor

1. When writing formulas, it may be cleaner to have one space at a time.

2. Wouldn't the cause of increase in prevalence include an increase in medical demand or an increase in people's interest?

Round 2

Reviewer 2 Report

Though I can't totally agree on the conclusion, but this article seems to have meaning as reference data with a large number of samples.